# The Modulatory Bioeffects of Pomegranate (*Punica granatum* L.) Polyphenols on Metabolic Disorders: Understanding Their Preventive Role against Metabolic Syndrome

**DOI:** 10.3390/nu15234879

**Published:** 2023-11-22

**Authors:** Mehdi Alami, Kaoutar Boumezough, Abdelouahed Khalil, Mhamed Ramchoun, Samira Boulbaroud, Tamas Fulop, Mojgan Morvaridzadeh, Hicham Berrougui

**Affiliations:** 1Department of Biology, Polydisciplinary Faculty, University Sultan Moulay Slimane, Beni Mellal 23020, Morocco; halamimehdi@gmail.com (M.A.); kaoutarrosa1@gmail.com (K.B.); ramchoun_10@yahoo.fr (M.R.); sboulbaroud@gmail.com (S.B.); 2Department of Medicine, Geriatrics Service, Faculty of Medicine and Biological Sciences, University of Sherbrooke, Sherbrooke, QC J1H 4N4, Canada; a.khalil@usherbrooke.ca (A.K.); tamas.fulop@usherbrooke.ca (T.F.); mojgan.morvaridzadeh@usherbrooke.ca (M.M.)

**Keywords:** *Punica granatum* L., pharmacokinetic, metabolic disorders, dyslipidaemia, diabetes mellitus, obesity, antihyperlipidemic, antihyperglycemic, atherosclerosis

## Abstract

Modern research achievements support the health-promoting effects of natural products and diets rich in polyphenols. Pomegranate (PG) (*Punica granatum* L.) contains a considerable number of bioactive compounds that exert a broad spectrum of beneficial biological activities, including antimicrobial, antidiabetic, antiobesity, and atheroprotective properties. In this context, the reviewed literature shows that PG intake might reduce insulin resistance, cytokine levels, redox gene expression, blood pressure elevation, vascular injuries, and lipoprotein oxidative modifications. The lipid parameter corrective capabilities of PG-ellagitannins have also been extensively reported to be significantly effective in reducing hyperlipidemia (TC, LDL-C, VLDL-C, and TAGs), while increasing plasma HDL-C concentrations and improving the TC/HDL-C and LDL-C/HDL-C ratios. The health benefits of pomegranate consumption seem to be acheived through the amelioration of adipose tissue endocrine function, fatty acid utilization, GLUT receptor expression, paraoxonase activity enhancement, and the modulation of PPAR and NF-κB. While the results from animal experiments are promising, human findings published in this field are inconsistent and are still limited in several aspects. The present review aims to discuss and provide a critical analysis of PG’s bioeffects on the components of metabolic syndrome, type-2 diabetes, obesity, and dyslipidemia, as well as on certain cardiovascular-related diseases. Additionally, a brief overview of the pharmacokinetic properties, safety, and bioavailability of PG-ellagitannins is included.

## 1. Introduction

The significant advancements in human history, particularly in the fields of transportation, food security, and telecommunications, have greatly improved the lives of 21st-century individuals. However, As a result of this materialistic evolution, people have become increasingly sedentary and are facing rising levels of stress. Concurrently, other unhealthy behaviours, such as sleep deprivation, smoking, alcoholism, and poor dietary habits, have exacerbated the situation. These lifestyle changes have had a profound impact on human health, potentially contributing to the development of numerous chronic illnesses, including cardio-metabolic and vascular diseases.

The preceding analysis partially explains the concerning prevalence rates observed for these diseases in recent years. In 2016, approximately 1.9 billion adults worldwide were diagnosed as overweight, ref. [1] with 650 million of them classified as obese. By June 2021, the same organization reported alarming percentages for overweight and obesity rates, standing at 39% and 13% of adults, respectively. Comparing these figures to those recorded in the mid-1970s, which were three times lower than today’s rates [1], it becomes evident that the world is facing a genuine risk of a global epidemic, given the exponential trajectory of future projections. Another factor that complicates the situation is the increasing prevalence of childhood obesity, which has become increasingly noticeable. This trend elevates the likelihood of individuals developing cardiovascular diseases, insulin resistance, psychological disorders, and ultimately, premature mortality.

According to the International Diabetes Federation [2], there were 415 million individuals diagnosed with diabetes in 2015. Future projections are deeply concerning, with the federation predicting an approximate rise to 642 million diabetics by the year 2040 [3]. From an economic perspective, the global financial burden of diabetes-related illnesses was estimated at around USD 1.31 trillion in 2015 [4]. These estimates encompassed all types of diabetes, spanning184 countries and considering adults aged between 20 and 79 years, along with direct and indirect expenditures. This burden is expected to escalate as the incidence of obesity continues to rise.

Pharmacological treatments, which include anti-obesity drugs, insulin-sensitizing agents, and insulin-secretagogue medications, have offered solutions to address these illnesses. However, pharmacovigilance reports and case series studies [5,6] have raised concerns about serious adverse effects associated with the use of some of these drugs. In this context, troglitazone was withdrawn from the US market in March 2000 [5], and the prescription of other similar drugs is significantly restricted and under current scrutiny [7], particularly due to concerns related to the risk of hypoglycaemia or coma [8,9,10], weight gain [11,12], cardiac events [13,14,15], hepatic dysfunction [5,16], and nephrotoxicity [17].

Over the past few decades, a substantial body of evidence has emerged to support the safety of natural nutrients and their significant connection to maintaining cellular homeostasis, preventing chronic diseases, enhancing cognitive function, and improving overall quality of life. Therefore, numerous research teams have directed their efforts towards identifying and characterizing the bioactive properties of natural compounds. Aromatic and medicinal plants, such as pomegranate (PG), have been reported in the ethnomedicine literature for their traditional use in treating microbial infections, diabetes mellitus, obesity, and related metabolic abnormalities. Currently, empirical evidence corroborates these traditional remedies and attributes potent normoglycemic and anti-dyslipidemic properties to PG consumption.

This review delves into the most relevant in vitro and in vivo data concerning the regulatory effects of PG consumption on hyperglycaemia and hyperlipidaemia. Additionally, We explore relevant clinical and pharmacokinetic studies conducted in this field.

## 2. Pharmacokinetic and Safety of Ellagitannin Constituents

Human studies on the bioavailability of ellagitannins (ETs) and ETs-derived nutrients are relatively recent, with the first paper on ETs from PG being published in 2004 [18]. Similarly, related animal research is also relatively recent and has primarily involved rats as an experimental model to explore the biotransformation, absorption, and clearance of ETs [19].

## 3. Catabolism of Ellagitannins

It has been established that under physiological conditions, ET polyphenols undergo non-enzymatic hydrolysis, involving acidic degradation and lysis byintestinal bacteria. In the colon, the resulting ellagic acid (EA) molecules undergo biochemical modifications, including dihydroxylation, decarboxylation, and lactone-ring cleavage, ultimately giving rise to dibenzopyran-6-one derivatives known as urolithins [20]. Notably, analysis of human faecal cultures has identified urolithin A as a product believed to be derived from EA and related polyphenols [20]. Importantly, the glucuronide or sulfate conjugates of this metabolite are not detected [20]. In a similar context, a study by Cerdá et al. [19] demonstrated that the intestinal microflora of rats had the capacity to transform ET-punicalagin into urolithin metabolites.

Furthermore, investigations conducted on jejunum and colon tissues [21] from Iberian pigs fed acorn ETs have demonstrated the presence of EA compounds and their bioconversion products. These studies also revealed inter-individual variability in the rate and profile of metabolites produced. This variability primarily stems from differences in the catabolic physiological capabilities, which depend on the composition and efficiency of gut microbial organisms and which can vary significantly among individuals.

## 4. Absorption and Bioavailability of Ellagitannins

Scientific experiments investigating the ability of PG compounds to enter human blood have indicated that EA can reach human plasma at a concentration (C max) of 31.9 ng/mL within 1 h of consuming 180 mL of PG juice containing 25 mg of EA. [18]. Similarly, a study conducted by Mertens-Talcott et al. [22] reported a C-max of 33 ng/mL for EA at a t(max) of 1 h following the consumption of a large amount (800 mg) of a standardized pomegranate extract. Additionally, In vivo studies have indicated that punicalagin can enter the bloodstream, which was detected at a C-max of 30 µg/mL in a group of rats supplemented with a diet containing 6% punicalagin [19]. However, polyphenols or related hydrolysis products from other sources, such as raspberries, were unable to enter the plasma circulation of healthy volunteers [23]. The production capacity of urolithins and their absorption capabilities exhibit significant variability, with numerous factors proposed to explain this condition. These factors include the physicochemical characteristics of the molecules (such as the chemical structure, degree of lipophilicity, solubility, etc.), and the individual-specific factors (such as microbiota composition, intestinal pH, etc.).

## 5. Biodistribution and Clearance of Ellagitannins

The accumulation of urolithin in body tissues appears to be notably concentrated in the colon and prostate organs [24]. However, the ability of ET metabolites, including punicalagin, to reach other organs, such as the liver and kidney, and to accumulate therein, is quite limited [19]. Moreover, some investigators have reported no detection of these metabolites [21]. The latter publication did not demonstrate any deposition of these molecules in the brain, adipose, or muscle tissues. The distribution and deposition capabilities of PG-ETs may be influenced by the biochemical characteristics of organ cells, such as their selective properties and permeability (e.g., in the case of the brain, the blood–brain barrier may restrict the entry of these molecules into the intraneuronal compartment), the overall structure of the ET-molecule (which may make it more or less susceptible to hepatic metabolism reactions), and individual variations in blood transport efficiency.

In terms of clearance, punicalagin metabolites have been detected in urine in the form of 6-H-dibenzo[b,d]pyran-6-one derivatives [19]. In fact, feces and urine investigations have been able to quantify approximately 3–6% of the total polyphenols ingested (ranging from 0.6 to 1.2 g daily). The authors have proposed the hypothesis that these compounds are almost entirely biotransformed into undetectable molecules or may possibly accumulate in unexamined tissues [19]. Furthermore, the clearance of ETs-polyphenols derived from other sources has also been studied. In an analysis following the ingestion of 35 g of walnuts, urolithin A was detected in excretions, with no detection of its sulfate or glucuronide derivatives [20]. However, another study confirmed the elimination of forms of urolithin glucuronide in feces [21], which aligns with the previously discussed inter-individual variability in terms of absorption rate, gutmicrobiota composition, and the degradation abilities of ETs by microbiota.

## 6. Safety of Pomegranate and Pomegranate Products

Although traditional remedy literature supports PG utilization and its potential to ameliorate health outcomes, some toxicological evaluations have indicated cellular component alteration and nuclear damage after PG administration. According to Tripathi and Singh [25], the concentration required to cause the death of 50% of snails of the species *Lymnaea acuminate* (LC50) using PG bark was determined to be 22.42 mg/L, and this value varied in a dose- and time-dependent manner. In addition, recombinogenic, mutagenic, and clastogenic effects have been observed in mice following the consumption of whole PG fruit [26], suggesting that PG may contain potentially toxic substances, most likely related to its alkaloid ingredients [25].

On the other hand, recent in vivo data [27], have been published, attributing antigenotoxic properties to PG fruit when administered at different concentrations (87.5, 175, 350, and 700 mg/kg of b.w). Additionally, 21 days of subacute toxicity assays did not reveal any toxicity markers. Moreover, an embryotoxicity study [28] showed that the safety of the hydroalcoholic extract of PG fruit at doses less than 0.1 mg per embryo, with the intraperitoneal administration lethal dose of 50 (LD50) only being reached at a high dose (731.1 mg/kg). Furthermore, Jahromi et al. [29], have suggested that the administration of the following doses of PG peel extract: 0.5, 1.9, and 7.5 mg/kg, did not provoke any toxic signs or behavioral disorder symptoms. Moreover, the data from this study did not indicate any disturbances in cholesterol, glucose, or hepatic enzyme levels.

Pure PG dietary compounds and their metabolites, such as punicalagin and EA, have also undergone investigation for their toxicity and safety properties [30,31]. In a 90-day subchronic experiment [30], rats from the F344 strain were provided fed a powdered basal diet containing 1.25%, 2.5%, and 5% doses of EA to assess its potential for altering biological processes.. The outcomes from this study did not show any toxic effects or clinical signs related to supplementation. Moreover, 37 days of repeated oral administration of a 6% ETs–punicalagin compound did not lead to liver or kidney toxicity [31]. Furthermore, this compound is not only safe, but also exhibits the ability to protect the hippocampal HT22 cells and H9c2 cardiomyocytes from toxicity induced by glutamate and doxorubicin [32,33]. In vitro and in vivo toxicological assessments of PG seed oil, a rich source of punicic acid, have concluded that this treatment is neither mutagenic nor clastogenic. Post-mortem analysis did not reveal any cellular abnormalities [34]. In this experiment, the concentration at which no adverse effects were observed was determined to be 4.3 g/kg/day.

Overall, the consumption of PG or pure PG compounds appears to be safe, and the doses at which toxic effects are expected to occur are significantly higher than those typically found in traditional ethnomedicine remedies and currently used for therapeutic purposes.

## 7. Pomegranate Consumption and Obesity

This pathology has a multifactorial etiology, which includes genetic, epigenetic, and environmental factors. Indeed, the unlimited access to a diet often rich in fat and lacking in fiber, combined with a considerable reduction of physical activity, in addition to the in-utero epigenetic modifications, constitute the primary elements underlying the obesity epidemic.

Physiopathological aspects of this energy imbalance are illustrated by the hormonal and metabolic changes that occur in the adipose tissue due to its exposure to a surplus of nutrients that exceeds the body’s energy expenditure. In fact, abdominal hypertrophy is accompanied by both qualitative and quantitative changes in the hormonal products produced by this endocrine organ. This is evident in the increased activation of pro-inflammatory genes (such as TNF-α, IL-6, etc.) and the reduced expression of mRNA and proteins related to insulin sensitivity, like adiponectin. Consequently, there is a concurrent state of chronic inflammation and hypo-adiponectinemia, which are associated with a sustained decrease in glucose uptake by myocytes, reduced fatty acid β-oxidation, and excessive glucose production. The activated form of fatty acids, acyl-COA, can activate a family of kinases (PKC), and through this pathway, the fatty acids participate in the enzymatic phosphorylation process of serine/threonine residues on insulin receptor substrates. This event disrupts insulin signalling transduction pathways (PI3 kinase and MAP kinase), contributing to the development of insulin resistance. The metabolic consequences of this pathological situation extend beyond the adipocyte realm, affecting the energy dynamics of the myocytes, hepatocytes, and the cellular homeostasis of all peripheral insulin-dependent tissues. 

The reversible nature of obesity allows for hygienic and dietary measures to restore metabolic homeostasis and in some cases, prevent the onset of other obesity-related pathologies. The nutritional aspect of this approach may be reinforced by improving the body’s antioxidant and anti-inflammatory status. This can be achieved through the consumption of vegetables and fruits with a low glycemic index, hydrogen atom transfer abilities, electron donor capacities, and the ability to chelate transition metals. The fruit of the PG has been suggested as a candidate for meeting these types of dietary requirements.

Experimental evidence attributed anti-obesity and anti-diabetic capabilities to the intake of PG (Table 1 and Table 2). According to Vroegrijk et al. [35], dietary supplementation with PG seed oil has shown its potency in reducing body weight and body fat mass, while increasing peripheral insulin sensitivity in C57Bl/J6 mice. These findings reinforce the previous observations made by Lei et al. [36], who observed a significant reduction in the main lipid parameters, including plasma total cholesterol (TC) concentration, triglyceride (TG) content, and TC/HDL-C ratio. Moreover, enzymes involved in lipid metabolism, such as acyl-CoA oxidase and carnitine palmitoyltransferase-1, as well as nuclear receptors like PPAR-α, have been positively influenced by PG flower constituents [37]. Furthermore, data from Oliveira et al. [38] showed a decrease in body weight after 30 days of feeding with PG extract, without any effect on plasma glucose concentration. However, contradictory findings concerning feed intake and weight gain tendencies have been published by Shabtay et al. [39]. Similarly, PG peel extract did not alter body weight gain and did not alleviate inflammation in an animal model of obesity (Balb/c mice) [40]. Nutrigenomic variability, in which individual genetic differences can influence the biological response to phytochemical nutrients, may explain the reported inconsistencies (Figure 1).

## 8. Type 2 Diabetes and Pomegranate Consumption

Various intrinsic factors, including genetic elements like specific nucleotide gene polymorphisms, as well as acquired factors stemming from environmental influences, such as dietary habits, in conjunction with the in utero epigenome changes (such as methylation, acetylation, etc.), play a determining role in the pathophysiology of insulin resistance. The intricate web of interconnected metabolic pathways enables this pathological mechanism to impact not only the onset of type 2 diabetes, but also cardiovascular and hepatic diseases like atherosclerosis and non-alcoholic fatty liver disease, respectively. The increase in circulating free fatty acids and serum pro-inflammatory cytokines (TNF-α, IL-6, etc.) that coincides with abdominal hypertrophy interfere with the insulin–substrate–receptor signaling cascade. This disruption prevent insulin’s signal transduction, thereby inhibiting its anabolic effects. On one hand, it leads to a decline in peripheral glucose uptake and glycogen synthesis. On the other hand, it results in the increased release of free fatty acids and improved hepatic glucose production. In response to the resulting hyperglycemia, the islets of Langerhans attempt to meet the growing insulin demand through the hyperinsulinism, temporarily restoring and maintaining cellular energy balance. However, the persistent insulin resistance gradually exhausts the β-insular function. Subsequently, the resultant state of glucolipotoxicity propels the shift from the non-pathological condition of glucose intolerance to the potentially harmful development of type 2 diabetes.

To prevent, or at least mitigate, such fatal consequences, it is advisable to consider natural molecules that feature an aromatic nucleus, along with hydroxyl groups. This is particularly pertinent when taking into account the potential toxic effects of anti-diabetic medications, as previously discussed. PG-polyphenols have garnered considerable attention in the realm of traditional medicine for their anti-diabetic effects. In fact, this plant may exert its effects through numerous mechanisms, including PPAR-γ activity modulation [37,58,75,78,79], resistin protein degradation [66], adiponectin gene expression [57], α-glucosidase enzymatic activities inhibition [59,63], Glut-4 mRNA expression [58], and β-mass regeneration [60]. 

In this context, Parmar et al. [59] reported that short-term treatment with PG peel extract can reduce α-amylase activity, serum glucose concentration, and lipid peroxidation content. The achieved hypoglycemic effects appear to be associated with an increase in insulin secretion capacities, as high levels of this hormone were found in the plasma compared to those of the control group. These results are consistent with those published by Li et al. [63]. This study demonstrated the ability of PG flowers to normalize postprandial hyperglycemia and to inhibit the catalytic activity of the α-amylase in a dose-dependent manner. Additionally, Vroegrijk et al. [35] demonstrated that PG seed oil treatment can improve peripheral insulin sensitivity in C57Bl/J6 mice. Furthermore, α-glucosidase and α-amylase assays showed the capacity of PG leaf extract to inhibit these enzymes, and there were also reports of enzymatic inhibition of pancreatic lipase [80]. However, there have also been contradictory findings regarding the anti-hyperglycemic potential of PG [40,61]. These studies did not indicate any significant changes following PG treatment. However, fasting blood glucose and pancreatic β-mass were not improved. 

PG preparations and pure PG-related constituents have also undergone investigation for their metabolic effects. These preparations include PG juice [81], punic acid [82], EA [83], punicalagin [84] and catalpic acid [67]. Outcomes from these studies suggest that these compounds exert numerous antidiabetic and biological effects.

PPAR-γ is a subtype of a nuclear receptor which is predominantly expressed in adipose tissue [85,86]. Its capability to influence physiological processes, such as adipocyte differentiation [86], lipid accumulation [87], lipoprotein lipase mRNA expression [88], pro-inflammatory gene repression, and resistin gene downregulation [89], have made this receptor a privileged target for anti-diabetic drugs. The genes under PPAR-γ control are believed to play a role in the development of insulin resistance and the progression of type 2 diabetes.

Phenolic molecules derived from PG may modulate PPAR-γ activities [37,58,75,78,79]. The observed antidiabetic effects resulting from this activation are believed to occur through two distinct pathways, depending on the ligand molecule involved. The first pathway involves selective agonist actions on PPAR-γ (without activatingthe associated PPAR-γ adipogenic factors like DRIP205/TRAP220), ultimately terminating with an improvement in insulin sensitivity [78]. This approach provides a solution to the weight gain frequently observed with the use of full activator antidiabetic drugs. The second pathway implies a different mechanism and consists of inhibiting PPAR-γ activation to prevent the associated undesirable adipogenesis effects. This activity has been attributed to quercetin, a flavonoid found in PGs [90,91]. Consequently, PG appears to exert a dual function, with its agonist PPAR-γ activities appearing to be weak in comparison to its PPAR-γ antagonistic actions [78]. In support of this perspective, Mueller and Jungbauer [78] showed that PG fruit extract (standardized to contain 40% EA) exhibited the highest PPAR-γ antagonism among fifty extracts screened for their PPAR-γ antagonistic activity. These findings regarding modulation are in line with those reported in in vitro [37,58,75] and in vivo [37,58,79] studies. Therefore, natural selective PPAR-γ agonists or in contrast, PPAR-γ antagonists, should be targeted as potential alternatives to thiazolidinediones in order to reduce adipose tissue hypertrophy.

Clinical studies conducted in this field have mainly utilized PG juice, and their outcomes have shown substantial inconsistentancy. In this regard, Parsaeyan et al. [92] have demonstrated that daily supplementation of 200 mL of PG juice for six weeks can reduce fasting blood sugar levels. Moreover, results from a systematic review of clinical studies concluded that pomegranate can be beneficial in reducing glycemia and improving insulin resistance [93]. However, unexpected findings suggest that PG juice did not modify the insulin secretory performance or hormone sensitivity [44]. This raises the possibility that the supplement may contain insulin-like compounds which can act through non-insulin mechanisms, especially given its capacity to alleviate inflammatory responses. Furthermore, arguments supported by statistical evidence emerge from a systematic literature review and meta-analysis of seven trials, indicating that PG intake did not show any significant effects on glycemic markers [94].

In summary, data from pure compounds seems to be consistent, whereas outcomes from PG extracts or products in both animal and human studies exhibit conflicting findings. This discrepancy can be explained by the probable interactions that could occur between different PG-bioactive substances contained in each extract or product, and these interactions have the potential to either enhance or diminish the cellular effect and the biological response (Figure 1).

## 9. Atheroprotective Activities and Antidyslipidemic Effects of Pomegranate Consumption

Vascular cell dysfunction and arterial endothelial tissue activation are believed to be the initial events in the atherogenic process. The increase in endothelial permeability observed in such circumstances allows LDL to migrate from the vascular compartment to the intima domain. In this region, these lipoproteins establish molecular interactions with matrix proteoglycans using their Apo-B100, and subsequently undergo post-translational biochemical modifications, in particular oxidative modifications. The local endothelial hypertrophy that occurs because of this non-physiological accumulation stimulates the recruitment of macrophages, specialized cells responsible for clearing cellular debris, through chemotactic signalling pathways. Simultaneously, complex phenomena involving a decrease in reverse cholesterol efflux capacity, migration and proliferation of smooth muscle cells, activation of metalloproteases, and the installation of a local pro-inflammatory secretory profile work in tandem to weaken the arterial wall, ultimately influencing the anatomical development and clinical expression of atherosclerosis. These series of harmful events are associated with dyslipidemia, visceral obesity, and diabetes mellitus.

These medical conditions are considered as major modifiable risk factors for atherosclerosis. Therefore, implementing safe strategies to rectify atherosclerosis-related metabolic abnormalities becomes crucial, as they can help restore vascular homeostasis. This may be accomplished through polymedication and lifestyle improvements. To achieve this goal, functional foods containing vitamins, phenols, and polyphenol compounds can be incorporated into this comprehensive approach. In this context, an extensive body of evidence has been accumulated to support the anti-atherogenic effects of a PG-based diet.

Poly-molecular substances found in PG exhibit a range of beneficial biological activities in the vascular microenvironment and can address several risk factors that impair the endothelial anatomy and function. Nutrients isolated from PG, such as ellagic acid, can suppress pro-matrix metalloproteinases and prevent their interaction with LDL molecules trapped in the arterial intima [95]. This contribution aids in inhibiting one of the first events involved in the pathogenesis of atherosclerosis. Moreover, compounds like punicalagin, gallic acid, and their derivatives are able to stimulate and enhance paraoxonase-1 activity in a HuH7 hepatocyte cells [96]. The preventive effects on atherosclerosis associated with PG consumption can also be attributed to improvements in eNOS expression [71] and the mRNA transcription of paraoxonase-1 [72,73,96].

Arterial protection mediated by PG substances may also occur through non-enzymatic mechanisms involving a significant reduction of VCAM-1, ICAM-1, and E-selectin molecules [41]. Furthermore, the TG/HDL-C ratio, a predictor factor for cardiovascular diseases [97], has shown a significant reduction after PG seed oil treatment [98]. These observations, along with others, suggest cardiomyocyte and coronary artery protection of PG supplementation. However, the effect on reverse cholesterol efflux appears unlikely, as investigators [41,48,69,92,99], did not find any significant amelioration in serum HDL-C concentration (Figure 2) and apolipoprotein-A levels, when compared to the placebo group.

## 10. Antidyslipidemic Effects of Pomegranate and Pomegranate Products

Hypercholesterolemia and hypertriglyceridemia are considered independent risk factors for chronic diseases like diabetes mellitus and atherosclerosis [102]. According to El-Hussieny et al. [76], PG peel and PG-ETs polyphenols have shown significant effectiveness in reducing hyperlipidemia (TC, LDL-C, VLDL-C, and TAGs), as well as improving plasma HDL-C concentrations in Wistar albino rats. Researchers have also examined these effects in human individuals [48,69,92]. Esmaillzadeh et al. [69] demonstrated that daily consumption of concentrated PG juice for 8 weeks can regulate lipid parameters in dyslipidemic individuals with type 2 diabetes. This regulation is primarily reflected in the decrease in serum TC and LDL-C concentrations, as well as the TC/HDL-C and LDL-C/HDL-C ratio. Similar results have been reported in Parsaeyan’s study [92], in which a short-term treatment (6 weeks) with 200 mL of PG juice resulted in a decrease in plasma TC and serum LDL-C levels. Identical results have been obtained in a medium-term supplementation (8 weeks), resulting in a significant reduction in plasma TC, LDL-C, LDL-C/HDL-C, and TC/HDL-C ratios [99]. However, contradictory outcomes have also been published, which did not suggest any changes in the lipid profile after PG intervention [48].

The lack of long-term supplementation and the relatively small number of participants are the main limitations of the reported clinical data. The latter concern has been addressed by a fixed-effect meta-analysis, which included 12 studies involving a total of 545 individuals [103]. Unfortunately, this analysis did not indicate any significant effect of PG intake or PG-derived products on the serum lipid profile. 

## 11. Insulin Resistance, Blood Pressure, and Pomegranate Consumption

The reduction in nitric oxide molecules, insulin resistance, oxidative stress, obesity, type 2 diabetes, and its associated vascular complications are believed to be the major factors leading to the development of vascular dysfunction and cardiovascular abnormalities. In these pathological conditions, the vasodilator effect and the performance of tubular sodium reabsorption, mediated by insulin hormones, are found to be altered. This situation is further exacerbated by the vasoconstrictive effects induced by the improper utilization of free fatty acids, contributing to blood pressure abnormalities.

Sufficient evidence from in vitro and in vivo studies [104,105,106,107] has prompted numerous research teams to investigate the antihypertensive properties of PG in various human conditions [41,46,47,48,72,108,109,110]. The outcomes of these clinical interventions show improvements in vascular function, the maintenance of lipid metabolism homeostasis, and the normalization of blood pressure. A systematic review and meta-analysis conducted by Sahebkar et al. [111] have confirmed these health-promoting capabilities. The authors suggest a significant beneficial impact of PG juice supplementation on both diastolic and systolic blood pressure, within a range of doses from 150 mL to 500 mL per day, containing polyphenol levels ranging from 0.7 mmol to 36.6 mmol, and for durations spanning from 4 weeks to 18 months. Moreover, additional evidence comes from a systematic review and meta-analysis published by Ghaemi et al. [112] supporting the beneficial effects of pomegranate consumption on both systolic and diastolic blood pressure. Furthermore, a non-randomized diet-controlled study reported a significant improvement in blood pressure in the group that consumed standardized extracts of pomegranate in comparison with that of the control group [113]. From a mechanistic perspective, the proposed operative actions of PG biochemical components include the inhibition of angiotensin-converting enzymatic activities [105,114] and the protection against tubular degeneration [106]. However, the included studies have limitations related to the number of individuals recruited [47,72], the lack of certain anthropometric measurements (such as age, sex, BMI) [46,47,72], and the absence of investigation into the main lipid parameters (LDL-C, HDL-C, TC, and TG) [47,72,109]. These limitations could provide a more comprehensive view of the health status of the enrolled individuals, thereby enabling a more reliable interpretation of the obtained results. 

## 12. Conclusions

The increasing prevalence of obesity significantly influences the incidence of obesity-related pathologies, such as non-alcoholic fatty liver disease, diabetes mellitus, and cardiovascular complications. Therefore, implementing measures to reduce and/or limit the progression of obesity and its pathological consequences is of crucial importance and should be a top public health priority. As we have reviewed, a considerable amount of research has been published, but the findings are often conflicting. This observed inconsistency can be attributed to various factors, including the choice of the plant part used, given the uneven distribution of the bioactive compounds, as well as factors like cultivar, geographical region, and their related bioclimatic and soil characteristics. Moreover, even if we were to standardize the fruits used, the results are sensitive to other variables related to the in vivo conditions, mainly plasma bioavailability, organ accessibility, and nutrigenomics considerations. All in all, there are substantial arguments supporting the medicinal value of PG and its bioactive substances in addressing the components of metabolic syndrome. The effective management of this cluster of metabolic abnormalities necessitates the adoption of comprehensive therapeutic strategies, with a central focus on hygienic-dietary recommendations and lifestyle modifications.

## Figures and Tables

**Figure 1 nutrients-15-04879-f001:**
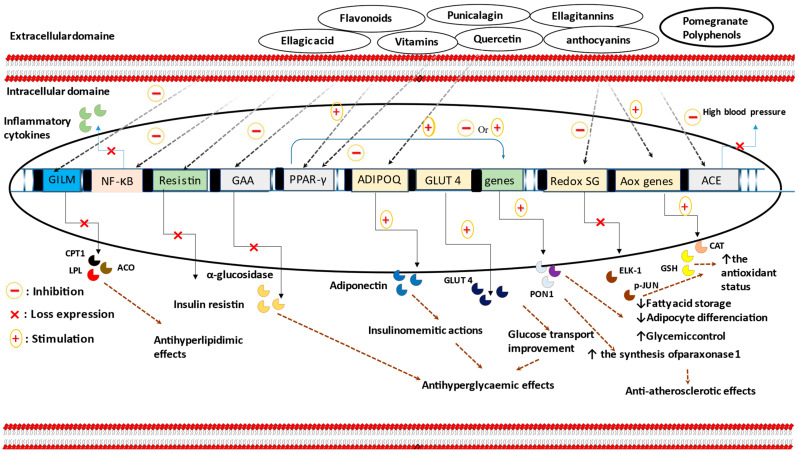
The Anti-hyperlipidemic and normoglycaemic molecular mechanisms of PG phytochemical active compounds. As summarized, PG nutrients are able to reduce resistin protein, α-glucosidase, and redox-sensitive gene expression, as well as increase adiponectinemia and ameliorate cellular glucose uptake. Inhibitory effects against enzymes involved in lipid metabolism, including carnitine palmitoyltransferase I, acyl-coenzyme A oxidase, and lipoprotein lipase, are also proposed. Furthermore, secondary metabolites from PG organisms could improve blood pressure, inflammatory state, and the antioxidant enzymatic capabilities, as well as modulate PPAR-γ activities. Abbreviations:—GILM: genes involved in lipid metabolism; CPT1: carnitine palmitoyltransferase 1; ACO: acyl-coenzyme A oxidase; LPL: lipoprotein lipase; redox SG: redox-sensitive genes; GAA: the gene that codes for α-glucosidase; AOX genes: antioxidant genes; CAT: catalase; GSH: glutathione; GLUT-4: glucose transporter-4; ADIPOQ: adiponectin gene; NF-κB: nuclear factor-kappa B; PPAR-γ: peroxisome proliferator-activated receptor gamma.

**Figure 2 nutrients-15-04879-f002:**
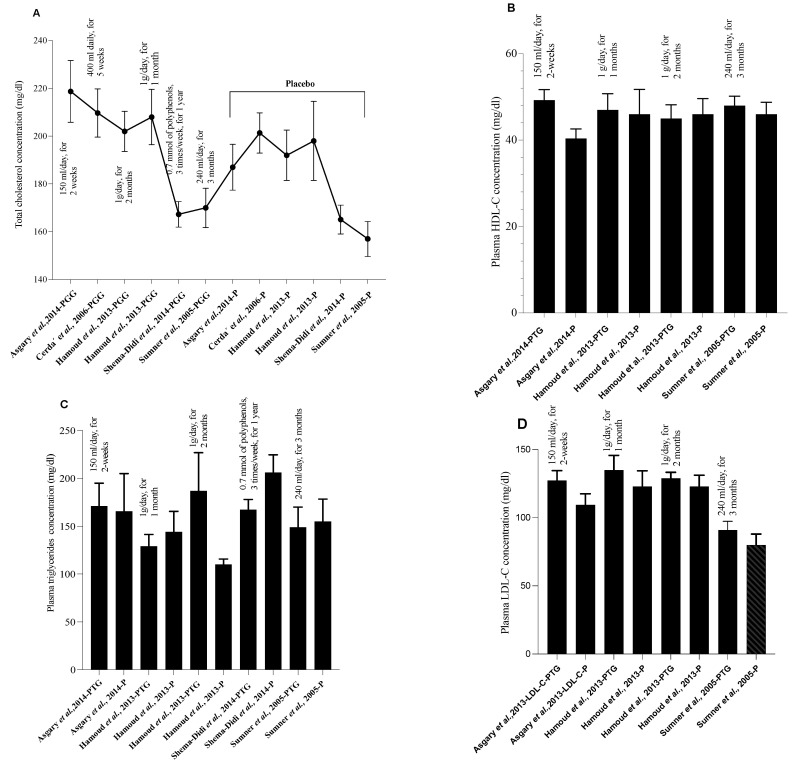
Pomegranate bioeffects on human lipid parameters. (**A**) Pomegranate consumption effect on total cholesterol [41,43,45,46,47]. (**B**) Pomegranate impac on plasma HDL-C concentration [41,45,47]. (**C**) Pomegranate intake and plasma triglycerides concentration [41,45,46,47]. (**D**) The modulatory effect of pomegranate supplementation on plasma LDL-cholesterol level [41,45,47]. (PTG: Pomegranate treatment group, P: placebo). All values are reported as the mean ± ESM. Pomegranate juice supplementation was also examined for possible atheroprotective effects. According to Burgermeister et al., PG juice products may modulate paraoxonase-2 (PON 2) activities through pathways dependent on PPAR-γ [100]. After one year of supplementation, it may be effective in increasing plasma paraoxonase activity and serum total antioxidant status by 83% and 130%, respectively [72]. Moreover, pomegranate intake may ameliorate lipid profile by reducing LDL-cholesterol and triglyceride levels [93]. PG juice treatment could also improve HDL-associated lactonase, paraoxonase-1 as well as arylesterase activities, when compared to the unsupplemented group [92,101]. This last finding is fundamentally importat because the enzymatic stability and biological activities of PON 1 improve when this enzyme is bound to HDL. Additionally, this preparation modulates paraoxonase-2 activity through PPAR-γ-related signaling cascades [100]. In in vivo experiments using apolipoprotein E-deficient mice, it was shown that when this product is added to the diet, it can increase macrophage cholesterol efflux and protect LDL against oxidative alterations [73], an initiating event and a crucial step in the atherosclerosis development. These bioprotective effects had been illustrated by attenuating LDL oxidation, macrophage cholesterol biosynthesis reduction [70], and redox-sensitive gene inactivation [71].

**Table 1 nutrients-15-04879-t001:** Summary of the most relevant human and animal findings related to the anti-hyperlipidemic effects of PG consumption.

Group	Population/Disease Induction	PG Part or Product	PG Dose and Duration	HDL	LDL	TC	VLDL	HDL-C	LDL-C	VLDL-C	TG	Refs.
**Clinical studies (the values are expressed as mean ± SD)**												
Treated group	Hypertensive individuals	PG juice	150 mL/day, for 2 weeks	ni	ni	218.73 ± 42.81 ^a^	ni	49.27 ± 8.06 ^a^	127.27 ± 24.22 ^a^	ni	171.18 ± 78.92 ^a^	[41]
Placebo group		-	-	ni	ni	187.00 ± 30.27 ^a^	ni	40.40 ± 6.91 ^a^	109.40 ± 25.82 ^a^	ni	165.60 ± 124.32 ^a^	
Treated group	Dyslipidemic patients	PG seed oil	800 mg twice daily, for 4 weeks	ni	ni	ni	ni	1.38 ± 0.44 ^b^	ni	ni	2.75 ± 1.40 ^b^	[42]
Placebo group		-	-	ni	ni	ni	ni	1.25 ± 0.26 ^b^	ni	ni	3.12 ± 1.59 ^b^	
Treated group	COPD	PG juice	400 mL daily, for 5 weeks	55.05 ±12.01 ^a^	130.48 ± 32.29 ^a^	209.68 ± 39.10 ^a^	ni	ni	ni	ni	170.68 ± 187.10 ^a^	[43]
Placebo group		-	-	56.75 ± 20.81 ^a^	116.06 ± 29.14 ^a^	201.34 ± 32.64 ^a^	ni	ni	ni	ni	137.91 ± 778.22 ^a^	
Treated group	Obese patients	PG juice	120 mL, for 1 month	ni	ni	4.7 ± 0.7 ^b^	ni	1.1 ± 0.1 ^b^	2.9 ± 0.8 ^b^	ni	1.3 ± 0.3 ^b^	[44]
Placebo group		-	-	ni	ni	4.8 ± 0.6 ^b^	ni	1.2 ± 0.2 ^b^	2.9 ± 0.5 ^b^	ni	1.2 ± 0.5 ^b^	
Treated group	Hyper-cholesterolemic patients	PGE +simvastatin	PGE: 1g/daySimvastatin: 20mg/day, for 2 months	ni	ni	202 ± 29 ^a^	ni	45 ± 11 ^a^	129 ± 15 ^a^	ni	187 ± 138 ^a^	[45]
Placebo group		-	-	ni	ni	192 ± 35 ^a^	ni	46 ± 12 ^a^	123 ± 27 ^a^	ni	110 ± 19 ^a^	
Treated group		PGE + simvastatin	PGE: 1g/day Simvastatin: 20 mg/day, for 1 month	ni	ni	208 ± 40 ^a^	ni	47 ± 13 ^a^	135 ± 37 ^a^	ni	129 ± 43 ^a^	
Placebo group		-	-	ni	ni	198 ± 55 ^a^	ni	46 ± 19 ^a^	123 ± 38 ^a^	ni	144 ± 71 ^a^	
Treated group	Hemodialysis patients	PG juice	0.7 mM of polyphenols, 3 times/week, for 1 year	36.8 ± 10.8 ^a^	100 ± 33.1 ^a^	167.3 ± 43.5 ^a^	ni	ni	ni	ni	167.3 ± 86.3 ^a^	[46]
Placebo group		-	-	34.3 ± 15.4 ^a^	94.3 ± 27.2 ^a^	165.1 ± 35.8 ^a^	ni	ni	ni	ni	206.1 ± 109.4 ^a^	
Treated group	CHD patients	PG juice	240 mL/day, for 3 months	48 ± 11 ^a^	91 ± 33 ^a^	170± 42 ^a^	ni	ni	ni	ni	149 ± 107 ^a^	[47]
Placebo group		-	-	46 ± 12 ^a^	80 ± 35 ^a^	157 ± 32 ^a^	ni	ni	ni	ni	155 ± 102 ^a^	
Treated group	Volunteers at high CVD risk	PG juice	500 mL/day, for 4 weeks	ni	ni	5.45 ± 1.0 ^b^	ni	1.52 ± 0.44 ^b^	3.31 ± 0.73 ^b^	ni	1.147 ± 0.39 ^b^	[48]
Placebo group		-	-	ni	ni	4.51 ± 0.51 ^b^	ni	1.46 ± 0.56 ^b^	2.54 ± 0.79 ^b^	ni	1.14 ± 0.51 ^b^	
**Animal studies (data is reported as mean ± SEM/SE/SD)**												
Treated group	Addition of 10% of lipid in the basal diet	Hydroethanolic extract of PG	50 mg/kg/day, for 23 days	89 ± 11 ^a^	209 ± 23 ^a^	87 ± 9 ^a^	ni	ni	ni	ni	381 ± 23 ^a^	[49]
			100 mg/kg/day, for 23 days	128 ± 5 ^a^	145 ± 29 ^a^	82 ± 5 ^a^	ni	ni	ni	ni	325 ± 43 ^a^	
			200 mg/kg/day, for 23 days	179 ± 18 ^a^	79 ± 8 ^a^	80 ± 9 ^a^	ni	ni	ni	ni	302 ± 31 ^a^	
			300 mg/kg/day, for 23 days	185 ± 20 ^a^	61 ± 7 ^a^	81 ± 7 ^a^	ni	ni	ni	ni	210 ± 27 ^a^	
Control+		-	-	98 ±9 ^a^	92 ±6 ^a^	73 ±8 ^a^	ni	ni	ni	ni	146 ± 21 ^a^	
Treated group	Hyper cholesterolemia diet	PG peel powder	(5%)	38.40 ± 5.18 ^a^	40.73 ± 1.85 ^a^	92.33 ± 4.76 ^a^	13.20 ± 0.69 ^a^	ni	ni	ni	66 ± 3.46 ^a^	[50]
			(10%)	36.93 ± 5.53 ^a^	46.67 ± 1.97 ^a^	96.00 ± 4.11 ^a^	12.40 ± 0.84 ^a^	ni	ni	ni	62± 3.69 ^a^	
			(15%)	41.50 ± 5.98 ^a^	45.77 ± 2.13 ^a^	97.67 ± 4.52^a^	10.40 ± 0.59 ^a^	ni	ni	ni	52 ± 3.99 ^a^	
		PG peel extract	(1%)	41.40 ± 5.18 ^a^	16.33 ± 1.85 ^a^	75.00± 3.66 ^a^	17.27 ± 0.79 ^a^	ni	ni	ni	86.33 ± 4.78 ^a^	
			(2%)	42.93 ± 5.53 ^a^	9.67 ± 1.97 ^a^	69.00± 3.89 ^a^	16.40 ± 0.67 ^a^	ni	ni	ni	82 ± 4.11 ^a^	
			(3%)	40.50 ± 5.98 ^a^	12.77 ± 2.13 ^a^	70.00± 3.92 ^a^	16.73 ± 0.64 ^a^	ni	ni	ni	83.67 ± 5.12 ^a^	
Control+		-	-	41.93 ± 5.53 ^a^	87.53 ± 1.97 ^a^	154.33 ± 5.13 ^a^	24.87 ± 0.77 ^a^	ni	ni	ni	124.33± 3.70 ^a^	
Treated group	High-fat diet	PG peel extract	200 mg/kg, for 56 days	ni	ni	172.3 ± 3.94	ni	40.03 ± 1.03	93.84 ± 3.69	38.49 ± 0.62	192.4 ± 3	[51]
Control+		-	-	ni	ni	271.8 ± 3.94	ni	29.30 ± 1.03	185.3 ± 3.69	57.26 ± 0.62	285.5 ± 3	
Treated group	Intraperitoneal injection of STZ (60 mg/kg)	PG flowers extract	250 mg/kg, for 21 dyas	ni	ni	124.50 ± 8.62 ^a^	ni	45.17 ± 4.84 ^a^	61.67 ± 6.12 ^a^	17.67 ± 4.50 ^a^	88.17 ± 7.05 ^a^	[52]
			500 mg/kg, for 21 days	ni	ni	118.67 ± 9.60 ^a^	ni	48.67 ± 5.16 ^a^	54.67 ± 4.89 ^a^	14.34 ± 2.95 ^a^	72.83 ± 6.52 ^a^	
Control+		-	0 mg/kg, for 21 days	ni	ni	292.33 ± 4.64 ^a^	ni	32.83 ± 4.22 ^a^	234.34 ± 6.12 ^a^	25.26 ± 0.93 ^a^	126.33 ± 4.64 ^a^	
Treated group	Intraperitoneal injection of STZ (65 mg/kg)	PG leaves	50 mg/kg, for 28 days	ni	ni	162.25 ± 5.28 ^c^	ni	37.79 ± 1.92 ^c^	106.22 ± 6.14 ^c^	18.76 ±0.73 ^c^	93.845 ± 3.66 ^c^	[53]
			100 mg/kg, for 28 days	ni	ni	142.38 ± 2.70 ^c^	ni	63.32 ± 3.11 ^c^	80.36 ± 2.08 ^c^	15.99 ± 1.57 ^c^	84.53 ± 4.49 ^c^	
			200 mg/kg, for 28 days	ni	ni	139.45 ± 1.98 ^c^	ni	44.54 ± 2.97 ^c^	60.41 ± 3.57 ^c^	15.71 ± 1.83 ^c^	76.25 ± 9.96 ^c^	
Control+		-	0 mg/kg, for 28 days	ni	ni	229.08 ± 7.51 ^c^	ni	20.47 ± 1.31 ^c^	179.50 ± 6.68 ^c^	29.09 ± 0.70 ^c^	145.46 ± 3.53 ^c^	
Treated group	Intraperitoneal injection of poloxamer 407	PG flowers	500 mg/kg, after 15 h	6.23 ± 0.39 ^b^	8.56 ± 0.62 ^b^	16.9 ± 0.60 ^b^	2.11 ± 0.23 ^b^	ni	ni	ni	10.57 ± 1.17 ^b^	[54]
Control+		-	-	5.04 ± 0.20 ^b^	9.9 ± 0.67 ^b^	18.39 ± 0.63 ^b^	3.38 ± 0.08 ^b^	ni	ni	ni	16.93 ± 0.75 ^b^	
Treated group		PG flowers	500 mg/kg, after 24 h	6.06 ± 0.29 ^b^	10.74 ± 0.95 ^b^	19.72 ± 0.67 ^b^	2.91 ± 0.09 ^b^	ni	ni	ni	14.56 ± 0.46 ^b^	
Control+		-	0 mg/kg	5.05 ± 0.17 ^b^	15.7 ± 0.80 ^b^	24.28 ± 0.89 ^b^	3.52 ± 0.09 ^b^	ni	ni	ni	17.66 ± 0.46 ^b^	
Treated group	High cholesterol diet	PG juice	0.2 mL/animal, for 30 days	78.58 ± 4.79 ^a^	19.38 ± 10.34 ^a^	135.83 ± 13.9 ^b^	37.87 ± 5.36 ^a^	ni	ni	ni	189.33 ± 26.81 ^a^	[55]
Control+		-		67.70 ± 2.34 ^a^	169.93± 31.90 ^a^	267 ± 31.78 ^a^	29.37 ± 1.18 ^a^	ni	ni	ni	146.83 ± 5.88 ^a^	
Treated group	High cholesterol diet	PGME	200 mg, for 30 day	1.54 ± 0.208	0.58 ± 0.118	1.93 ± 0.191	0.23 ± 0.06	ni	ni	ni	1.05 ± 0.17	[56]
			300 mg, for 30 day	1.29 ± 0.68	0.21 ± 0.057	1.63 ± 0.125	0.10 ± 0.028	ni	ni	ni	0.91 ± 0.12	
			400 mg, for 30 day	0.91 ± 0.117	0.17 ± 0.049	1.04 ±0.159	0.07 ± 0.026	ni	ni	ni	0.46 ± 0.15	
Control+		-	0 mg	2.16 ±0.150	0.68 ± 0.050	2.55 ± 0.211	0.27 ± 0.072	ni	ni	ni	1.17 ± 0.13	

Abbreviations. TC: total cholesterol; TG: triglycerides; HDL: high-density lipoprotein; HDL-C: high-density lipoprotein-cholesterol; LDL: low-density lipoprotein; LDL-C: low-density lipoprotein-cholesterol; VLDL: very low-density lipoprotein; VLDL-C: very low-density lipoprotein cholesterol; ni: not investigated; ^a^: mg/dL; ^b^: mmol/L; COPD: chronic obstructive pulmonary disorder; PGME: PG mesocarp extract; CHD: ischemic coronary heart disease; ^c^: (mg %); CVD: cardiovascular diseases; COPD: chronic obstructive pulmonary disease; STZ: streptozotocin.

**Table 2 nutrients-15-04879-t002:** Summary of the main effects of PG intake on type 2 diabetes, obesity, and cardiovascular diseases.

Animal Model/ Population/Cell Line	Disease or Induction of the Disease	PG Part or Product	Dose and Period of Treatment	Findings	Refs.
**PG intake effect on obesity and diabetes mellitus**					
Male C57Bl/J6 mice	High-fat diet	PG seed oil	1%, for 12 weeks	↓ body weight; ↓ body fat mass; ↔ liver insulin sensitivity; ↑ peripheral insulin sensitivity; ↔ food intake; ↔ energy expenditure.	[35]
Male CD-1 mice	High-fat diet	PG seed oil	61.79 mg/day, for 14 weeks	↓ weight gain; ↓ body weight; ↓ absolute weight gain; ↓ percentage of weight gain; ↔ lean mass; ↔ cholesterol profile; ↓ leptin; ↑ adiponectin.	[57]
Zucker diabetic fatty rats	Genetic manipulation	PG flower extract	500 mg/kg/day, for 6-weeks	↓ plasma glucose; ↔ fasting serum glucose; ↑ cardiac PPAR-γ mRNA expression; ↑ GLUT-4 mRNA; ↑ mRNA expression of inhibitor-kBα.	[58]
Zucker lean rats	-			↔ plasma glucose; ↔ fasting plasma glucose.	
Human THP-1-derived macrophage cells	-		50 µg/mL, for 48 h	↑ PPAR-γ gene expression; ↑ PPAR-γ-dependent mRNA expression. ↑ lipoprotein lipase activity.	
Swiss albino male mice	Alloxan injection	PG peel extract	200 mg/kg/day, for 10 days	↓ plasma glucose; ↓ α-amylase activity; ↓ water consumption; ↓ lipid peroxidation; ↑ plasma insulin.	[59]
Male albino rats	Alloxan injection	PG peel aqueous extract	0.43 g/kg BW, for 4-weeks	↓ blood glucose; ↑ insulin level; ↑ β-cells regeneration.	[60]
Male Sprague Dawley rats	Alloxan injection	PG seed	60 g/kg/day, for 15 days	↔ serum glucose; ↔ fasting blood glucose; ↔ size of islets; ↔ islets density; ↔ percent of β-cells in each islet; ↔ number of islets.	[61]
albino rats	Alloxan injection	PG flower extract	300 or 400 or 500 mg/kg. Sampling time: at 1 and 2 h.	↓ blood glucose.	[62]
Zucker diabetic fatty rats	Genetic manipulation	PG flower extract	250,500, and 1000 mg/kg/day, for 2 weeks. 200 µL, for 5 min for the in vitro assay.	↓ postprandial hyperglycemia; ↓ α-glucosidase activity (IC50: 1.8 µg/mL); ↓ plasma glucose levels after sucrose loading.	[63]
Adult albino rats	Streptozotocin treatment	PG seed extract	150, 300 and 600 mg/kg, for 2, 4, 8 and 12 h	↓ blood glucose in time and dose-dependent manner.	[64]
3T3-L1 pre-adipocytes	-	Punicic acid	1.25, 2.5, 5 and 10 µM	↑ PPAR-α and γ activity; ↓ fasting plasma glucose; ↑ glucose normalizing capabilities; ↓ NF-κB activation; ↓ TNF-α expression.	[65]
Ovariectomized mice	Surgical intervention	PG fruit extract	30 mg/kg/day, for 12 weeks	↓ serum resistin concentrations.	[66]
3T3-L1 adipocytes	-		50 and 100 µg/mL, for 9 and 12 h	↓ resistin protein secretion; ↔ resistin mRNA expression; ↑ intracellular resistin degradation; ↔ adiponectin secretion.	
		Ellagic acid	20, 40, and 70 µM, for 12 h	↓ resistin protein secretion; ↔ adiponectin secretion; ↓ intracellular resistin time-dependently.	
		Punicic acid	5 and 10 µM, for 9 h	↔ resistin molecule secretion.	
C57BL/6J obese mice	High-fat diet	Catalpic acid	1g/100g, for 78 days	↓ insulin; ↓ fasting blood glucose; ↑ glucose normalizing ability; ↓ abdominal white adipose tissue storage; ↑ PPAR-α expression; ↑ HDL-C; ↓ TG.	[67]
Male and female ICR mice	High-fat diet	PG leaf extract	400 or 800 mg/kg daily, for 5 weeks	↓ body weight; ↓ energy intake; ↓ TC; ↓ TG; ↓ TC/HDL-C ratio; ↓ glucose; ↓ fat absorption; ↓ appetite.	[36]
Zucker diabetic fatty rats	Genetic manipulation	PG flower extract	500 mg/ kg daily, for 6 weeks	↓ TG; ↓ TC; ↓ fatty acids; ↓ fatty acids transport proteins; ↓ PPAR-; ↓ acyl-CoA oxidase; ↓ 5 -AMP-activated protein kinase-α-2; ↓ carnitine palmitoyltransferase-1; ↓ acetyl-CoA carboxylase mRNA.	[68]
Zucker diabetic fatty rats	Genetic manipulation	PG flower extract	500 mg/kg/day, for 6 weeks	↓ TG; ↓ lipid droplets; ↑ PPAR-α; ↑ acyl-CoA oxidase; ↑ carnitine palmitoyltransferase-1; ↓ gene expression of stearoyl-CoA desaturase-1; ↔ fatty acids and TG synthesis; ↔ fatty acids and TG hydrolysis; ↔ fatty acids and TG uptake.	[37]
HepG2 cell line	-		10, 50 and 100 µg/mL, for 48 h	↑ PPAR-α; ↑ Acyl-CoA oxidase mRNA.	
Type 2 diabetic and hyperlipidemic patients	Diabetes mellitus and hyperlipidemia	PG juice	40 g/day of concentrated PG juice, for 8 weeks	↓ TC; ↓ LDL-C; ↓ LDL-C/HDL-C; ↓ TC/HDL-C; ↔TG; ↔ HDL-C.	[69]
Calves	-	Polyphenols PG extract	5 or 10 g/day (0,15, and 30 mg of gallic acid equiv/kg/day), for 70 days	↔ on body weight or intake, during the first 30 postnatal days, but are ↓ after this period; ↔ glucose concentration; ↔ 3-hydroxybutyrate; ↓ fat digestion; ↔ dry matter; ↔ starch digestibility; ↔ organic matter.	[38]
Calves	-	PG peel	Ad libitum, for 2 months	↑ feed intake; ↑ weight gain tendency.	[39]
Balb/c mice	High-fat diet	PG peel extract	0.2% (6 mg/day/mouse), for 4 weeks	↔ weight gain; ↔ glycaemia; ↔ glucose tolerance; ↓ TC; ↓ LDL-C; ↔ IL-1β, IL-6 and COX-2 in the liver; ↓ IL-1β, IL-6 and COX-2 both in in the gastrointestinal tract and visceral adipose tissue.	[40]
Male Wistar rats	High-lipid diet	PG peel extract	50, 100, 200, and 300 mg/kg, for 23 days	↓ body weight; ↓ TC; ↓ LDL-C; ↓ alkaline phosphatase; ↓ TG; ↑ HDL-C; ↓ AST; ↓ ALT.	[49]
**Pomegranate intake and cardiovascular diseases**					
J774.A1 macrophages	-	PG juice	75 mmol/L, for 90 min	↑ Ox-LDL degradation by 40%; ↔ on macrophage degradation of native LDL; ↔ macrophage cholesterol efflux capacities; ↓ macrophage cholesterol biosynthesis (by 50%).	[70]
Human coronary artery endothelial cells	High shear stress	PG juice	7–14 µL of PG juice, for 24 h.	↓ the activation of redox-sensitive genes (ELK-1 and p-JUN); ↑ eNOS expression.	[71]
Low-density-lipoprotein receptor-deficient mice (LDLR−/− mice)	Genetic manipulation and high-cholesterol diet	PG juice	31 µL/day (0.875 µmol of total polyphenols), for 24 weeks	↓ the activation of redox-sensitive genes (ELK-1 and p-JUN); ↑ eNOS expression; ↓ the progression of atherosclerosis lesions in mice.	[71]
Carotid artery stenosis individuals	carotid artery stenosis	PG juice	50 mL, for 1 or 3 years	↓ carotid intima-media thickness; ↑ PON 1 activity; ↓ LDL basal oxidative state; ↓ LDL susceptibility to oxidation; ↓ antibodies against ox-LDL; ↑ total antioxidant status; ↓ antibodies against oxidized LDL; ↓ systolic blood pressure.	[72]
Apolipoprotein E-deficient mice	Genetic manipulation	PG juice	31 µL of PJ/d (0.875 mmol of total polyphenols/d), for 2 months	↑ PON1 activity; ↓ MPM lipid peroxide; ↓ Ox-LDL MPM uptake; ↓ MPM cholesterol esterification; ↑ macrophage cholesterol efflux; ↓ macrophage Ox-LDL uptake; ↓ cholesterol esterification; ↓ atherosclerosis lesions size.	[73]
Apolipoprotein E-deficient mice	Genetic manipulation	PG byproduct	17 or 51.5 µg of gallic acid equiv/kg/day, for 3 months	↓ atherosclerotic lesion size; ↓ cellular lipid peroxide; ↓ glutathione levels; ↑ POM-2 lactonase activity; ↓ Ox-LDL uptake.	[74]
J774A.1 macrophage	-		10 or 50 µmol/L of total polyphenols, for 18 h	↓ cellular total peroxide; ↓ Ox-LDL uptake.	
J774A.1 macrophage	-	PG juice	10–50 µM of total polyphenols, for 18 h	↑ expression and enzymatic activity of PON-2; ↑ PPAR-γ and AP-1 activation; ↓ macrophage oxidative status; ↓ Ox-LDL uptake.	[75]
Wistar albino rats	High-fat diet	PG peel extract	50 or 100 mg/kg, for 6 weeks	↓ TC; ↓ LDL-C; ↓ VLDL-C; ↓ TAGs; ↑ HDL-C; ↑ GR; ↑ SOD; ↑ CAT; ↑ GSH; ↓ MDA; ↑ PON-1 activities; ↓ LDH activity; ↑ TNF-α; ↑ CD36.	[76]
		Ellagic acid	1 mg/kg, for 6 weeks		
		Punicalagin	7 mg/kg, for 6 weeks		
Zucker diabetic fatty rats	Genetic manipulation	PGF extract	500 mg/kg, for 6 weeks	↓ cardiac fibronectin expression; ↓ collagen I and III mRNAs; ↓ expression of endothelin -1; ↓ endothelin receptor a; ↓ c-jun and inhibitor-kBβ expression; ↑ inhibitor-kBα; ↓ LPS-induced NF-kB activation.	[77]

Abbreviations. ↔: no effect; ↓: decrease; ↑: increase; TC: total cholesterol; HDL-C: high-density lipoprotein cholesterol; LDL-C: low-density lipoprotein cholesterol; TG: triglycerides; VLDL: very low-density lipoprotein; VLDL-C: very low-density lipoprotein cholesterol; TAGs: triacylglycerols; ox-LDL: oxidized LDL; LDLR_/_: low-density-lipoprotein receptor-deficient; CAS: atherosclerotic patients with carotid artery stenosis; MPM: mice peritoneal macrophages; COX-2: cyclooxygenase-2; TGCOX-2: transgenic mice COX-2; PPAR: peroxisome proliferator-activated receptor; NF-κB: nuclear factor-kappa B; LPS: lipopolysaccharide; PON: paraoxonase; eNOS: endothelial nitric oxide synthase; TNF-α: tumor necrosis factor α; ALT: alanine transaminase; AST: aspartate transaminase; GLUT: glucose transporter; GSH: glutathione; SOD: superoxyde dismutase; GR: glutathione reductase; CAT: catalase; CD36; cluster of differentiation 36; MDA: malondialdehyde; AP1: activator protein 1.

## Data Availability

Not applicable.

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
