# Peer review of "The Modulatory Bioeffects of Pomegranate (Punica granatum L.) Polyphenols on Metabolic Disorders: Understanding Their Preventive Role against Metabolic Syndrome"

_nutrients, 2023, doi:10.3390/nu15234879_

Round 1

Reviewer 1 Report

Comments and Suggestions for Authors

It is confused about “suggest the bioavailability of EA at the following concentrations: 31.9 ng/mL and 33 ng/mL (Line 106-107)” and “ET-punicalagin molecules were detected at a concentration of 30 μg/mL, in a rat animal model(Line 109-110).” What on earth is the bioavailability of EA or ETs based on existing data? What is the meaning on 31.9 ng/mL, 33 ng/mL and 30 μg/mL? How many ellagitannins was consumed? What factors contributed the great difference from ng/mL to μg/mL besides ET catabolism into EA?

LC50 of PG bark on snail Lymnaea acuminate was 22.42 mg/l according to Ref 25. How about the values of PG/EA on rats or mice with mg/kg.bw. (LD50)?

With respect to anti-obesity, anti-hyperlipidemic capacities, etc., seed oil, juice/extract/peel extract, and flower constituents of PG, all exerted the beneficial effect despite some opposite findings. What were the main functional constituents from different parts of PG? How about their concertation, especially ETs?

Author Response

Response to reviewer 1
Thank you very much for your valuable comments that helped us to improve this work. The manuscript has been much improved, and we think that all your comments have been taken into a consideration.

Comment 1: “It is confused about “suggest the bioavailability of EA at the following concentrations: 31.9 ng/mL and 33 ng/mL (Line 106-107)” and “ET-punicalagin molecules were detected at a concentration of 30 μg/mL, in a rat animal model (Line 109-110).”
Response: We have rephrased the mentioned sentences. We hope they are now clear.

Comment 2: “What on earth is the bioavailability of EA or ETs based on existing data?” 
Response: The aim is to show that PG's bioactive molecules, such as ellagic acid and punicalagin, which have shown interesting anti-inflammatory and antioxidant capacities in vitro, can reach the blood circulation, and thus exert a biological effect comparable to the effects obtained in vitro. Based on the cited publications, it can be said that these molecules reach the plasma and can therefore be studied for their possible biological activities in in vivo. 

Comment 3: “What is the meaning on 31.9 ng/mL, 33 ng/mL and 30 μg/mL?” 
Response: The meaning of these values is: the maximum concentration (C-max) of the compounds in plasma, after consumption of an initial amount of PG product. The initial quantity follows intestinal transformations and may reach the blood circulation or may not. The mentioned studies represent a proof for the bioavailability of the cited PG-compounds. 

Comment 3: How many ellagitannins was consumed?
Response: In the form of PG-juice, or in the form of the commercial diet containing desired amount of ET-polyphenols, or as a capsule. For more details, please refer to the materials and methods section of the cited articles.
Comment 4: "What factors contributed the great difference from ng/mL to μg/mL besides ET catabolism into EA?”
Response: There is many influencing factors that can reduce or increase their plasma concentration:
1.    Factors related to individuals participated to the study such as intestinal absorption rat, metabolic activities across gastrointestinal tract, microbiota composition, intestinal pH, etc.
2.    Factors related to the compound/physicochemical characteristics of molecule: lipophilicity, solubility, and chemical structure, etc.
3.    Factors related to the co-ingested diet and technique used for quantifying the C-max.
Comment 5: LC50 of PG bark on snail Lymnaea acuminate was 22.42 mg/l according to Ref 25. How about the values of PG/EA on rats or mice with mg/kg.bw. (LD50)?
Response: 
The LD50 of PG extract on mice was: 731,1 mg/kg. The LD50 of PG extract on rat is not reported by the authors. However, authors suggest 0.4 and 1.2 mg/kg to be safe. 
For EA the LD50 was not reported. However, the authors reported the no-observed-effect level, and it was: 3011 mg/kg b.w./day for males, and 3254 mg/kg b.w./day for females.

Reviewer 2 Report

Comments and Suggestions for Authors

Dear Editor,

I carefully read the manuscript "The Modulatory Bioeffects of Pomegranate (Punica granatum L.) Polyphenols on Metabolic Disorders: Understanding their Preventive Role against Metabolic Syndrome".

My comments and suggestions for the authors are the following:

 - 

 - Pag. 20: "for doses ranging from 150 ml to 500 ml per day". The authors should more properly refer to the concentration of the active ingredients rather than the fruit juice volume.

 - The authors should specify how the data searching process was conducted.

 - English language is rough and very difficul to read. The language needs to be carefully revised and improved.

 - Pag. 20: "come to a conclusion" should be replaced by "comes to a conclusion".

- Pag. 20: "[89][47][107][108][48][109][41][46]". References should be included in the manuscript following the Instructions for the Authors of the Journal.

 - Pag. 20: "Gender" should be replaced by "sex".

 - Pag. 20: "Cardiovascular abnormalities" should be replaced by "cardiovascular complications".

 - Throughout the manuscript, the authors replace the word "subjects" with "individuals".

 - The authors should high consider to refer to doi: 10.3390/nu14081665, doi: 10.1002/ptr.7952 and doi:10.3390/app12199673.

Comments on the Quality of English Language

Please, see my comments above.

Author Response

Cover Letter

Dear Editor,

We thank you and the reviewers for your valuable time in reviewing our article. We appreciate your helpful comments and observations that led to possible improvements in the current version. The authors have carefully considered the comments and have done their best to respond to each one. We have included the point-by-point responses below. All changes in the manuscript were made using the "Track Changes function" to facilitate detection of each modification introduced in this new version. The authors welcome additional constructive feedback, if necessary.

Sincerely,

Response to reviewer 2

Thank you for your valuable observations and comments.

Comment 1: Pag. 20: "for doses ranging from 150 ml to 500 ml per day". The authors should more properly refer to the concentration of the active ingredients rather than the fruit juice volume.

Response: Done (page 20, highlighted in yellow)

Comment 2: The authors should specify how the data searching process was conducted.

Response:  As this work is a literature review and not a meta-analysis or systematic review, we have not followed a search strategy. However, we used PubMed, EMBASE, MEDLINE, google scholar, and Clinical Trials.gov databases to search, using potential keywords related to the research field, such as "Punica granatum L.” AND “Metabolic Syndrome"; "Punica granatum L.” AND “Obesity" "Punica granatum L.” AND “Diabetes mellitus " "Punica granatum L.” AND “Atherosclerosis or dyslipidemia".

"Punica granatum L. or Pomegranate” AND “Metabolic desorders” .

 Comment 3:  English language is rough and very difficul to read. The language needs to be carefully revised and improved.

Response: The current version has been carefully reread by our team and checked by a colleague fluent in English writing. We hope that enhance the linguistic quality of our paper.

Comment 4: Pag. 20: "come to a conclusion" should be replaced by "comes to a conclusion".

Response: Done

Comment 5: Pag. 20: "[89][47][107][108][48][109][41][46]". References should be included in the manuscript following the Instructions for the Authors of the Journal.

Response: Done. All the successive references have been merged in this way [ R1,R2] for two successive references R1 and R2. Similarly for the sequence which includes more than two successive references: example [ R1] [R2] [R3] [R4], they become like this [R1-R4], as indicated by the author's instructions of journal.

Comment 6: - Pag. 20: "Gender" should be replaced by "sex".

Response: Done

Comment 7: - Pag. 20: "Cardiovascular abnormalities" should be replaced by "cardiovascular complications"

Response: Done

Comment 8: Throughout the manuscript, the authors replace the word "subjects" with "individuals".

Response: Done

Comment 9: - The authors should high consider to refer to doi: 10.3390/nu14081665, doi: 10.1002/ptr.7952 and doi:10.3390/app12199673.

Response: Done. We revised and included the mentioned paper in our analysis. We cited them in appropriate section and context.

Round 2

Reviewer 2 Report

Comments and Suggestions for Authors

Dear Editor,

I carefully read the revised version of the manuscript, that is significantly improved compared to its original version.